# Validating Domain-Specific Cytology Pretraining for Thyroid FNAC Classification under Limited-Data Conditions

**Anthony Dreier**                          ADREIER2413@FLORIDAPOLY.EDU
**Lien Tran**                                      LIEN@REIINC.ORG
**Luan Nguyen**                              DRLUAN.ENT@GMAIL.COM
**Alex Mremi**                                   ALEXMREMI@GMAIL.COM
**Danielle Range**                          DANIELLE.RANGE@DUKE.EDU
**Walter Lee**                                   WALTER.LEE@DUKE.EDU
**Hoan T. Ngo**                                  HNGO@FLORIDAPOLY.EDU
*Corresponding author: Hoan T. Ngo, PhD, Florida Polytechnic University, Lakeland, FL-USA*

## Abstract

Developing models for thyroid cancer detection using fine-needle aspiration cytology (FNAC) images is challenging due to data scarcity. We present a controlled comparison between CytoFM, a cytology-specific foundation model, and a strong ConvNeXt-base baseline under a matched multiple-instance learning protocol. Using 5-fold cross-validation, 4 random seeds, and a frozen test set with pooled out-of-fold calibration, CytoFM achieved a test PR-AUC of 0.959 and ROC-AUC of 0.850, compared to 0.952 and 0.833 for ConvNeXt-base. Although limited by a small cohort, the results suggest that domain-specific cytology pretraining yields better score separation on the held-out test set under limited-data smartphone microscopy conditions.

**Keywords:** thyroid FNAC, cytology foundation model, multiple instance learning, smartphone microscopy, domain-specific pretraining

## 1. Introduction

Thyroid fine-needle aspiration cytology (FNAC) is an established, cost-effective tool for evaluating thyroid nodules, yet slide digitization and expertise are often unavailable in resource-limited settings. Smartphone microscopy offers a low-cost alternative, and prior work has shown that these images can support machine learning analysis (Assaad et al., 2023).

Recent pathology foundation models focus primarily on histology (Chen et al., 2024; Lu et al., 2024; Zimmermann et al., 2024). However, cytology differs significantly in morphology and acquisition, suggesting a generic backbone may not be optimal. We address this by employing CytoFM, a recently developed cytology-specific foundation model (Ivezic et al., 2025), and compare with ConvNeXt-base (Liu et al., 2022) in a matched evaluation pipeline.

## 2. Methods

### 2.1. Cohort and preprocessing

We used the NOH thyroid FNAC cohort from the National ENT Hospital in Vietnam (Assaad et al., 2023). From 132 patients (2,017 smartphone-captured images), we excluded

scan NOH062 due to significant imaging artifacts and five nondiagnostic cases, leaving 126 patients. We used a patient-level split: 88 training, 19 validation, and 19 frozen test bags (one bag per patient). The frozen test set (15 malignant, 4 benign) was used only for final evaluation.

Images were converted to RGB, cropped to the largest non-black contour, resized to $224 \times 224$, and normalized with ImageNet statistics. Standard training augmentations (flips, rotation, color jitter, blur) were used; evaluation was deterministic.

## 2.2. Backbone comparison and MIL model

We compared two backbones: CytoFM (a ViT-Base encoder pretrained with iBOT (Zhou et al., 2022)) and ConvNeXt-base with generic weights. Per-image embeddings were aggregated using a gated-attention multiple-instance learning head (Ilse et al., 2018). CytoFM used a short frozen warm-up followed by fine-tuning; ConvNeXt-base remained frozen after preliminary tests showed no gain from unfreezing.

## 2.3. Training and evaluation

Training used stratified 5-fold cross-validation across 4 seeds (42, 123, 777, 2024). We optimized weighted cross-entropy with early stopping on validation PR-AUC. Thresholds were selected by pooling out-of-fold validation predictions to target 0.90 specificity. This conservative operating point prioritized high specificity to minimize false-positive malignant calls. PR-AUC and ROC-AUC were primary metrics; sensitivity and specificity served as secondary indicators.

## 3. Results

Table 1 summarizes the results. While ConvNeXt-base showed slightly higher mean cross-validation performance, CytoFM achieved the stronger calibrated test results (PR-AUC 0.959 vs 0.952; ROC-AUC 0.850 vs 0.833). This suggests that while the generic backbone fits the training distribution well, the cytology-specific backbone offers stronger held-out performance on the frozen test set.

Table 1: Patient-level 4-seed comparison. Cross-validation (CV) values are mean±SD. Test values are final results after pooled out-of-fold threshold calibration.

| Model | CV PR | CV ROC | Test PR | Test ROC | Sens. | Spec. |
|---|---|---|---|---|---|---|
| CytoFM | 0.910±0.048 | 0.735±0.098 | **0.959** | **0.850** | **0.600** | 0.750 |
| ConvNeXt-base | **0.931**±0.035 | **0.759**±0.105 | 0.952 | 0.833 | 0.533 | 0.750 |

At the calibrated operating point, CytoFM detected 9/15 malignant cases (sensitivity 0.600) versus 8/15 for ConvNeXt-base (0.533), both with 3/4 benign cases correctly identified. Because the frozen test set contains only four benign cases, a single misclassification inherently alters specificity by 25%. Consequently, the one-case difference in sensitivity (9/15 vs 8/15) is somewhat more informative here than specificity alone at this operating point. Figure 1 highlights the qualitative difference: CytoFM achieved a broader

spread of calibrated probabilities, placing benign cases farther below the threshold, whereas ConvNeXt-base predictions clustered near the boundary.

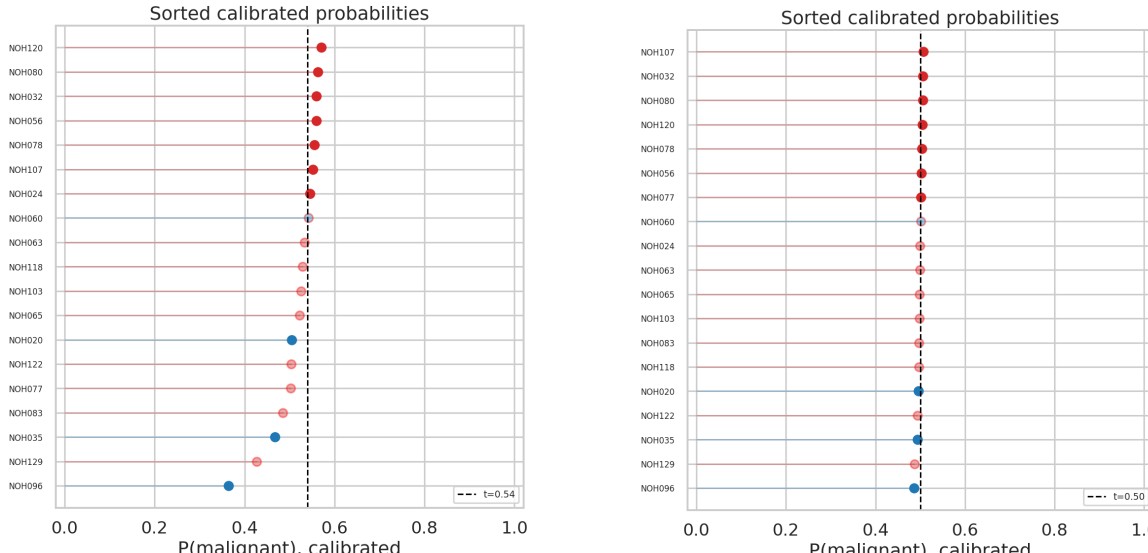

Figure 1: Sorted calibrated probabilities on the frozen test set. CytoFM (left) demonstrates better score separation between classes, while ConvNeXt-base (right) shows higher uncertainty near the decision threshold.

## 4. Discussion and Conclusion

Domain-specific cytology pretraining provides measurable value even in limited-data FNAC scenarios. The divergence between cross-validation and test performance is particularly notable; it suggests that while a generic backbone may easily fit the training distribution, the cytology-specific backbone may transfer more effectively to the held-out smartphone test set. Although the test set is small, the consistent improvement in AUC and score separation suggests that CytoFM may learn more transferable features for smartphone-captured cytology than generic pathology backbones. Future work should validate these findings on larger cohorts, but this study supports the continued development of cytology-specific foundation models for low-cost digital pathology.

## Acknowledgments

Acknowledging NIH funding (1R21CA268428-01, 4R33CA268428-03) and the clinical team for data collection.

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
