# OpenReview forum: "Validating Domain-Specific Cytology Pretraining for Thyroid FNAC Classification under Limited-Data Conditions"
_MIDL.io/2026/Short_Papers — MIDL 2026 - Short Papers Poster_

### Official Review · Reviewer_e4jG · 2026-05-03
**Interesting study but very limited scope and results**

**Rating:** 2
**Confidence:** 4

**Review:**

The results are interesting but not particularly suprising or novel, and the overall study is too small to be considered as an actual validation study.

**Summary:**

This paper studies classification performance of smartphone-based cytology images, from different backbones: one pre-trained on general histopathological data, and another one specializing to histopathology.

The results, when tested on a private test set (19 samples), showed consistently better results with the cytology backbone.

**Strengths:**

- Relevant task and addressing both domain-gap and "resource-gap", when looking at data from places with less resouces (e.g., smartphone-based scans
- Evaluation done on a private test set that is unlikely to have made it to the training set of the foundational models used

**Weaknesses:**

- The size of the dataset is quite limited (132 patients)
- A single task is evaluated
- Data comes from a single hospital, making it difficult to assess if the results are a "fluke" or a generalizable pattern

**Justification Of Rating:**

The results are interesting but not particularly suprising or novel, and the overall study is too small to be considered as an actual validation study.

---

### Decision · Program_Chairs · 2026-05-08

Accept (Poster)